# Tourism Village Development: Measuring the Effectiveness of the Success of Village Development

Fafurida Fafurida [1,2,*], Yunastiti Purwaningsih [1], Mulyanto Mulyanto [1] and Suryanto Suryanto [1]

1 Faculty of Economics and Business, Universitas Sebelas Maret, Surakarta 57126, Indonesia
2 Faculty of Economics, Universitas Negeri Semarang, Semarang 50229, Indonesia
* Correspondence: fafurida@mail.unnes.ac.id

**Abstract:** The objectives of this research are to identify the distribution of tourist villages in the Central Java Province using a spatial approach and to analyze the current conditions regarding: (a) the comparison of the conditions of tourist villages and non-tourist villages from the input aspects (attractiveness, accessibility, amenity) supporting tourism development, (b) the comparison of the conditions of tourist villages and non-tourist villages seen from the output of tourist village development achievements (Sustainable Development Goals (SDGs) and the Developing Village Index (DVI)), and (c) the influence of input factors on output factors with the status of tourist villages as a moderating variable. This research seeks to examine the area of villages in the Central Java Province, with 8562 villages focusing on 384 tourist villages and 8178 non-tourist villages. The analytical methods used to answer the objectives of this research are Geographic Information System (GIS), Mann–Whitney test (U test), and Partial Least Square (PLS) analysis. The existence of tourist villages in general can have a positive impact in increasing attractiveness, amenities, and SDGs. These findings prove that the existence of a tourist village is able to support the achievement of the Village SDGs. This is in accordance with previous studies, which found that tourism development can have a positive effect on village economic conditions. In terms of accessibility, there is no significant difference between tourist villages and non-tourism villages because currently the development of accessibility infrastructure in rural areas is evenly distributed, not only in tourist villages but in all villages. As for DVI, the existence of tourist villages has not been able to encourage the achievement of DVI because there are many elements that make up DVI not only aspects of attractiveness, accessibility and amenities but also elements of social, economic and environmental resilience.

**Keywords:** tourist villages; development; spatial; sustainable development goals; developing village index

## 1. Introduction

The tourism sector plays a significant role in the world economy, contributing to economic growth, job creation, and sustainable development. The key role of the tourism sector is economic contribution. The tourism sector is a significant contributor to the global economy. The tourism sector generates revenue through tourist spending on accommodation, food, transportation, souvenirs and other tourism-related activities. According to the World Travel and Tourism Council (WTTC), the global travel and tourism industry contributed 10.4% to world GDP in 2019 and supports more than 330 million jobs. The second key role is job creation: the tourism sector is a major source of employment, particularly in developing countries. The sector provides jobs in a variety of fields, including hospitality, transportation, retail and entertainment. WTTC estimates that the tourism sector accounts for one in ten jobs worldwide. Third, tourism promotes cultural exchanges between people from different countries, helping to foster mutual understanding and respect. It provides an opportunity for travelers to learn about the history, customs and traditions of different cultures. Fourth, the tourism sector can also contribute to efforts to protect and preserve the

environment. Sustainable tourism practices, such as ecotourism, can help conserve natural resources and protect ecosystems while reducing negative impacts on the environment. Fifth, the tourism sector can encourage infrastructure development in various fields, such as transportation, communication, and accommodation. This can help improve the quality of life of local people and attract further investment in the area. Overall, the tourism sector plays an important role in the global economy.

Indonesia is a country that has great potential in the tourism sector. The tourism sector is one of the priority development sectors of the Indonesian government at this time. Tourism has an important role in increasing economic growth. Tourism development is also a stimulus for investors to invest in the infrastructure sector to support tourism (Sakai 2006). Rational and balanced policies in the tourism development sector can be one of the main factors in successfully overcoming the crisis trend in rural areas in many countries (González-Ramiro et al. 2016).

The tourism sector has a role in socioeconomic development. With strategic investments adapted to the local context, tourism can really contribute to sustainable growth (Ibanescu et al. 2018). Tourism also contributes to increasing employment opportunities and increasing income (Camelia 2014). When tourism development occurs in one region, it may encourage the emergence of job opportunities for the surrounding community due to the business potential that comes along with tourism development. With these job opportunities, the income of people around tourism development will increase. This is also strengthened by Nizar (2015) in his study in Indonesia, which analyzed the effect of the number of tourists and tourism foreign exchange on the economic growth, where the results show that tourism affects economic growth.

The commitment in tourism development must at least meet the aspects of tourism development, which certainly will have a positive impact on the condition of infrastructure, economic, and social facilities in the region where the tourism development is conducted. To support the performance of the tourism sector, various development programs can also support the achievement of the Sustainable Development Goals (SDGs) such as villages without poverty and hunger, economic villages that grow evenly, villages that care for the environment, villages that are culturally responsive, etc. Further improvement of these conditions will certainly affect the status of regions or villages to become independent, developed, or underdeveloped according to the Developing Village Index (DVI) criteria. DVI functions to determine the status of village progress and independence, providing basic data and information for village development.

Much of the literature and several field studies show that the tourism industry in a region is considered to have a positive impact on improving the economy (C.-H. Chin et al. 2014; Giannakis 2014; Chuang 2011; Dimitrovski et al. 2012; Fatimah 2015; Fong and Lo 2015; Anuar et al. 2013; Ibanescu et al. 2018; Lee 2013; Ritchie 2004; Teodoro et al. 2017; Villanueva-álvaro et al. 2017). The tourism industry is one of the fastest growing sectors in the world economy in recent years. The contribution of the total gross domestic product (GDP) obtained from travel and tourism around the world in 2000–2021 shows that travel and tourism contributed 6.1 percent of the global gross domestic product (GDP) in 2021. This shows an increase compared to last 2020 but it remains below the number reported before the coronavirus (COVID-19) pandemic. Overall, the total contribution of travel and tourism to the global GDP will amount to around USD 5.81 trillion in 2021 (World Travel and Tourism Council 2021).

Some developing countries have made tourism the main star in international trade and the main source of foreign exchange, including Indonesia. The Indonesian tourism industry has the potential to increase foreign exchange reserves, because this country has extraordinary natural beauty and has a variety of cultural products in the form of temples, dances, customs, and many other unique things in every region and even in all villages in Indonesia. Tourism wealth will not decrease along with more foreign tourists visiting and enjoying their holidays in a country. Therefore, tourism plays an important role as a profitable export commodity for the country. Not only that, the tourism industry is also a

major driver for socioeconomic growth by providing new jobs, business opportunities, and infrastructure development.

In many regions, the government develops tourist villages that in the future are expected to have a high positive effect on the village and the surrounding area. This is in accordance with a research by Škufli and Štoković (2011), which states that the tourism industry can be a driving force in overcoming the economic problems in a region. Tourist villages and various related activities can be considered as the main axis of a rural development strategy that can protect a fragile socioeconomic order in the short term and create sustainable development in the long term (Giannakis 2014). The development of tourist villages is considered to be able to improve the community's economy.

Central Java is one of the provinces in Indonesia with high tourism potential. Central Java has a diversity of different and unique tourism, and it also has aspects of availability of tourism facilities and infrastructure such as the availability of complete transportation, including the existence of airports, ports, railroads, and terminals, in addition to the number of accommodations and inns that are spread in various regions. Central Java is one of Indonesia's tourism destinations that offers many charming natural, cultural, and artificial tourist attractions. Government policy encourages the creation of tourist villages in each regency/city, which can be seen from the significant increase in the number of tourist villages in the last six years. In 2015, the number of tourist villages in the Central Java Province was 126, and it increased every year to 717 tourist villages in 2021.

As an effort to overcome the problem of inequality between urban and rural areas, the potential for developing tourist villages is one of the instruments that can be used as a means of increasing the economy of rural areas. Based on the number of distributions, the largest tourist villages are in the Semarang Regency with a total of 55 tourist villages. The distribution of the number of tourist villages in the Central Java Province is presented in Figure 1.

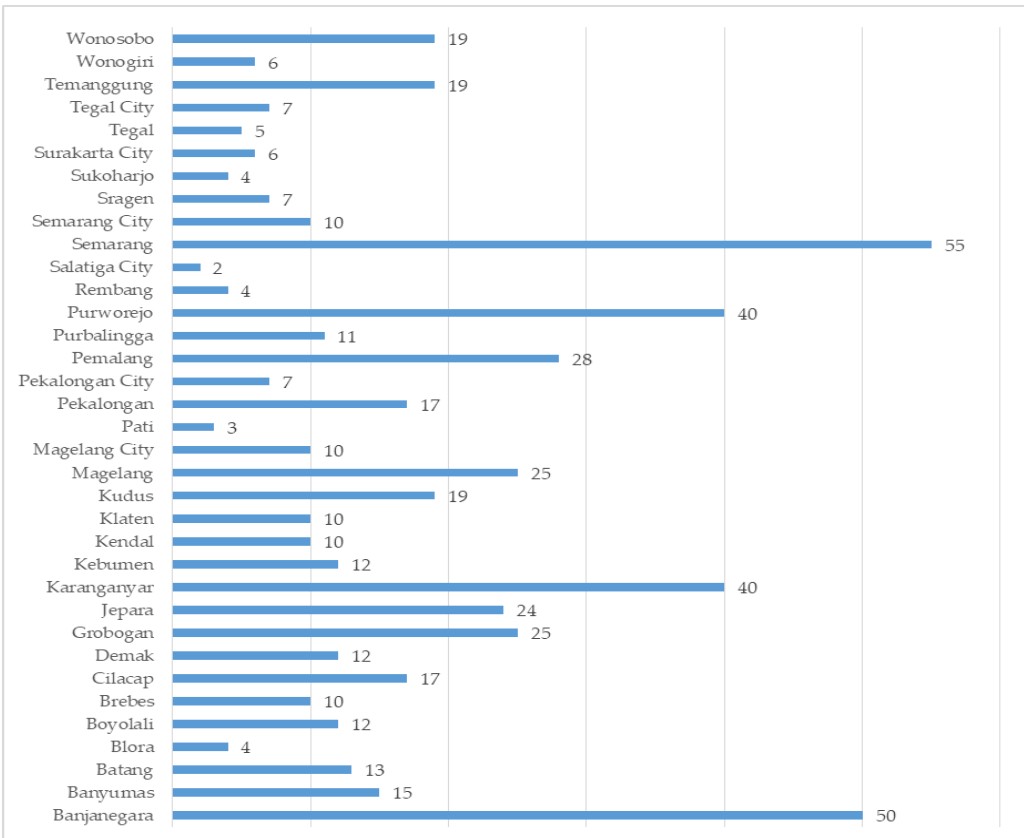

**Figure 1.** Number of Tourist Villages in Central Java Province based on Regencies/Cities in 2021.

It can be seen from Figure 1 that all regencies or cities in Central Java have tourist villages. In 2021, the highest number of tourist villages were in the Semarang Regency with 55 tourist villages, followed by the Banjarnegara Regency with 50 tourist villages and the Purworejo Regency and Karanganyar Regency with the same number of tourist villages, 40. The area with the lowest number of tourist villages is Salatiga City with a total of two tourist villages. The high number of tourist villages in all districts/cities in Central Java Province shows that the potential for village tourism in Central Java is very high.

This research will try to identify the distribution of tourist villages in Central Java Province using a spatial approach and to analyze the current conditions regarding: (a) the comparison of the conditions of tourist villages and non-tourist villages seen from the input aspects (attractiveness, accessibility, amenity) supporting tourism development, (b) the comparison of the conditions of tourist villages and non-tourist villages seen from the output of tourist village development achievements (SDGs and the Developing Village Index), and (c) the influence of input factors on output factors with the status of tourist villages as a moderating variable. Here the output of this research is expected to be able to provide a clear picture of the current conditions of the existing tourist village and to be able to evaluate how it affects the condition of the village seen from the output factors, the SDGs, and the Developing Village Index (DVI).

## 2. Methodology

This research is a combination of quantitative and qualitative research. The quantitative approach in this research has been used to identify differences in the input factors (attractiveness, accessibility, and amenity) and the output factors (SDGs and DVI) in tourist village areas and non-tourist villages, and it has also been used to analyze the influence of input factors on output factors with the status of a tourist village. The quantitative approach that has been used in this case is explanatory, which is conducted by giving an objective on an explanation of the causes and effects of a particular notion, idea, phenomenon, or symptoms. By contrast, the qualitative approach in this research has been used to identify the distribution of tourist villages in Central Java Province, Indonesia.

This research examines the area of villages in the Central Java Province, with 8562 villages focusing on 384 tourist villages and 8178 non-tourist villages. This research has also examined the input factors for the development of tourist villages, which are the attractiveness, accessibility, and amenity of each village with 36 indicators. In addition, this research examines the output factors for the development of tourist villages with the SDGs and Development Village Index (DVI) variables. The type of data used in this research is secondary data. The secondary data are obtained from various sources, the Indonesian Central Bureau of Statistics (BPS), and the Tourism Office of Central Java Province.

The analytical method that has been used to answer the objectives of this research is the Geographic Information System (GIS), which is used to answer the first objectives, identifying the distribution of tourist villages. Furthermore, analyzing the differences in the conditions of the attractiveness, accessibility, and amenity from tourist villages and non-tourist villages was conducted through an average different test analysis. The inferential statistic used is the Mann–Whitney test (U test). The analysis used next is Partial Least Square (PLS), which in this research is used to determine how far the input factors of the tourist village, consisting of the aspects of the attractiveness, accessibility, and amenity aspects, affect the output factors of tourist villages, which consist of the Village Sustainable Development Goals (SDGs) and the Developing Village Index (DVI).

The structural equation model contained in this research is formulated based on the proposed hypothesis. This research has five main variables and one control variable, in which three of the five variables, namely attractiveness, accessibility, and amenity, are exogenous latent constructs (independent variables), and the next two variables, Village SDGs and Developing Village Index, are endogenous latent constructs (dependent variable). The moderating variable in this research is the status of a tourist village. The model

equation for the status variable of a tourist village as a moderating variable in this research is as follows:

$$Y_1 = \beta_1 X1 + \beta_2 X2 + \beta_3 X3 + \beta_4 ZX1 + \beta_5 ZX2 + \beta_6 ZX3 + e$$

$$Y_2 = \beta_1 X1 + \beta_2 X2 + \beta_3 X3 + \beta_4 ZX1 + \beta_5 ZX2 + \beta_6 ZX3 + e$$

Description:

$Y_1$: SDGs
$Y_2$: DVI
Z: Tourist Village Status
X1: Attractiveness
X2: Accessibility
X3: Amenity
$\beta_1 \ldots \beta_6$: Variable coefficient
*e*: *error*

The Partial Least Squares (PLS) model showing the relationship between the input and output factors being tested can be seen in Figure 2.

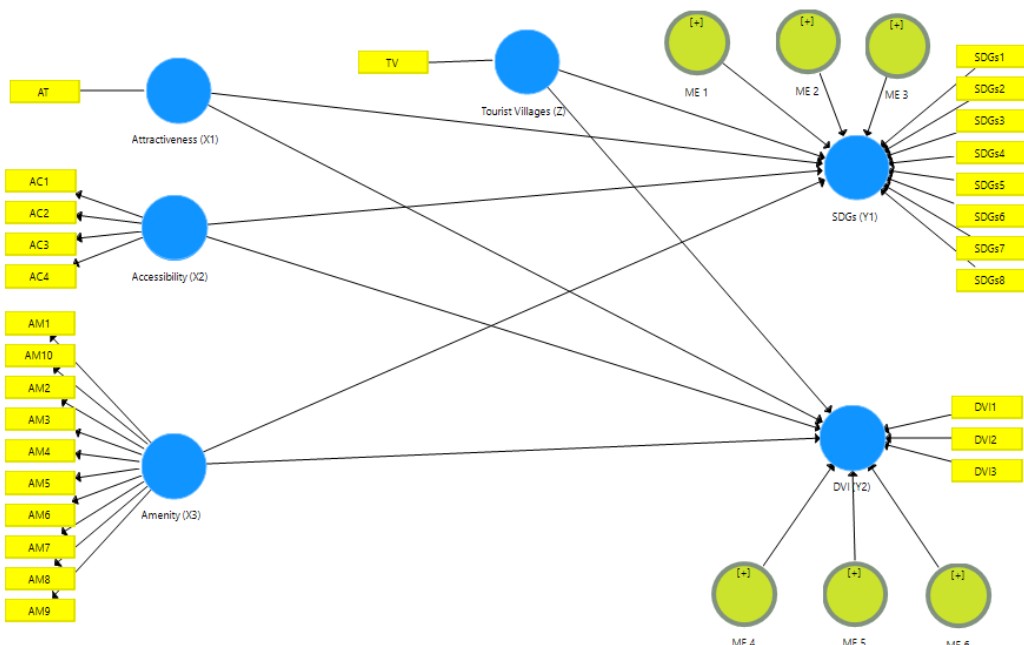

**Figure 2.** Partial Least Squares (PLS) Design Analysis with Tourist Village as Moderating Variable.

The model in this research is the Partial Least Square (PLS) using three input latent variables with a reflective model and two output latent variables with a formative model. The input variables in the research model are the attractiveness variable (X1) with one indicator, the accessibility variable (X2) with four indicators, and amenity variable (X3) with ten indicators, while the output variables are the SDGs (Y1) with eight indicators and the DVI variable (Y2) with three indicators (Y2). The model in this research also has a moderating variable, which is the status of a tourist village (Z). An explanation of the variables that have been used in the Partial Least Squares (PLS) analysis design can be seen in the following table.

## 3. Results and Discussion

The effort to photograph the distribution of tourist villages in this research uses the Geographic Information System (GIS) tool to see the spatial distribution. From the data obtained from the Tourism Office of Central Java Province, the latest data in 2021 show that Central Java has 384 tourist villages and 1069 tourist attractions. In the systematics of

work in attracting tourists, the attractiveness in the form of tourist villages and non-tourist villages usually support each other. The more attractiveness that exists in an area, the more tourists will come to visit it. From the number of tourist village and non-tourist village attractiveness spread over the Central Java Province, Figure 3 shows the results of the mapping conducted using the Geographic Information System (GIS) to see the distribution patterns of tourist village locations and tourist attractiveness in Central Java Province.

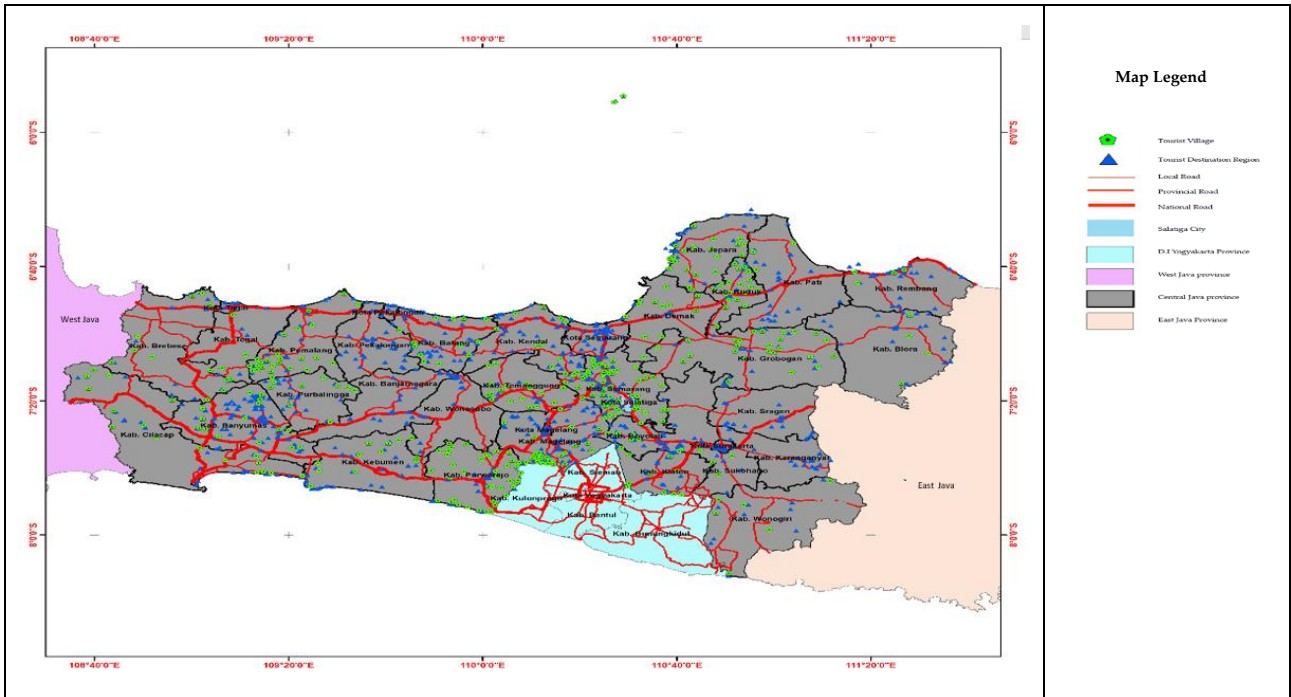

**Figure 3.** Distribution Mapping of Tourist Village and Tourist Destination Region in Central Java Province.

From the results of mapping the distribution of tourist village locations and tourist attractions in Central Java, as presented in Figure 3, it can be seen that spatially there are groupings of tourist village location points in some regions. This grouping is found in several areas in the Semarang Regency, Kudus Regency, Magelang Regency, Purworejo Regency, and Pemalang Regency. This grouping pattern shows that tourist villages are developing in regency areas that have high enough number of villages, have rural tourist attractiveness, and have good infrastructure.

The same thing is also found in the previous studies such as those conducted by (Calado et al. 2011) in the Azores, which found that rural tourism potential tends to focus on coastal areas, in accordance with the existing rural tourism facilities. The areas identified is in accordance with the best supporting infrastructure locations. Another study that is relevant to this finding is that conducted by Xi et al. (2015) in the tourist area of Yesanpo, who found that the emergence of a "core-periphery" pattern of spatial evolution of rural settlements closer to the point of tourist attraction, where the level of intensity of land used shows higher density, which leads to increased tourism and change landscape pattern. The results of this study are also in line with the research conducted by Hardati (2019), and González-Ramiro et al. (2016), where by using the same analysis the distribution of tourist villages was determined and the patterns of grouping were identified.

By using the Geographic Information System (GIS), the accessibility of each tourist village location can also be identified properly. Accordingly, determining the development of the existing tourist villages can in the future be based on the latest existing conditions based on this research.

Furthermore, to find out whether there is a significant average difference between variables in the tourist village group and the non-tourist village group, Table 1 offers the following results:

**Table 1.** Grouping of Variables Based on the Outer Model.

| | Reflective | | | Formative | |
|---|---|---|---|---|---|
| | Faktor Input | | | Faktor Output | |
| Variable | Code | Indicator | Variable | Code | Indicator |
| Attractiveness | DT | Open public space | | SDGs 1 | A village without poverty and hunger |
| Accessibility | AK 1 | The widest type of road surface | | SDGs 2 | The village economy grows evenly |
| | AK 2 | The widest type of road surface | | SDGs 3 | Health care village |
| | AK 3 | Public transport that passes through the village: Presence of public transport | | SDGs 4 | Environmentally friendly village |
| | AK 4 | Public transport through the village: Main public transport operations | SDGs | SDGs 5 | The village cares about education |
| Amenity | AM 1 | Internet presence | | SDGs 6 | Women friendly village |
| | | Cell phone/mobile phone signal | | | |
| | | Cellphone/cell phone internet signal | | | |
| | AM 2 | Number of Base Transceiver Station (BTS) towers | | SDGs 7 | Networked village |
| | | Number of cellular telephone communication service operators | | SDGs 8 | Cultural responsive village |
| | AM 3 | Number of mosques | | DVI 1 | Social Resilience Index (IKS) |
| | | Number of prayer rooms | | DVI 2 | Economic Resilience Index (IKE) |
| | | Number of Christian churches | | | |
| | | Number of catholic churches | | | |
| | | Number of chapels | | | |
| | | Number of pura | | | |
| | | Number of monasteries | | | |
| | | Number of kelenteng | Indeks Desa Membangun (DVI) | | |
| | | Number of other places of worship | | DVI 3 | Environmental Resilience Index (IKL) |
| | AM 4 | Number of government commercial banks | | | |
| | | Number of private banks | | | |
| | AM 5 | Number of shopping groups | | | |
| | | Number of minimarkets/ supermarkets | | | |
| | | Number of shops/grocery stalls | | | |

**Table 1.** *Cont.*

| Reflective | | | Formative | | |
|---|---|---|---|---|---|
| **Faktor Input** | | | **Faktor Output** | | |
| **Variable** | **Code** | **Indicator** | **Variable** | **Code** | **Indicator** |
| | AM 6 | Number of markets with permanent buildings | | | |
| | | Number of markets with semi-permanent buildings | | | |
| | | Number of markets without buildings | | | |
| | AM 7 | Number of restaurants/eateries | | | |
| | | Number of food stalls/shops | | | |
| | AM 8 | Number of hotels | | | |
| | | Number of accommodations: hostel/motel/inn/guesthouse | | | |
| | AM 9 | Number of hospitals | | | |
| | | Number of health centers with hospitalization | | | |
| | | Number of puskesmas without hospitalization | | | |
| | | Number of auxiliary health centers | | | |
| | | Number of polyclinics/clinics | | | |
| | AM 10 | Number of pharmacies | | | |
| | | Number of drugstores/herbs | | | |

Based on Table 2, it can be seen that from the variables analyzed, almost all have a significant average difference between the tourist village group and the non-tourist village group because the two-way (t-tailed) significance value is <0.05; only the accessibility variable has no significant difference because the significance value is 0.439 > 0.05. The results of this research are in accordance with a similar study conducted in Romania by Ibanescu et al. (2018), in which there is a significant difference between rural areas developed by the tourism sector that are visited by tourists and those that are not. By using the same analysis, the Mann–Whitney U test, this research also demonstrates that there is a significant positive influence between villages with tourist visits and those without tourist visits, which is indicated by a higher value of all the indices analyzed.

**Table 2.** Mann–Whitney U Test Result of Tourist Village and Non-Tourist Village on Input Variables.

| | **Test Statistics** [a] | | |
|---|---|---|---|
| | **DT** | **AK** | **AM** |
| Mann–Whitney U | 1,262,348.500 | 1,530,518.000 | 1,341,863.500 |
| Wilcoxon W | 1,336,268.500 | 34,974,449.000 | 34,785,794.500 |
| Z | −7.403 | −0.937 | −4.823 |
| Asymp. Sig. (2-tailed) | 0.000 | 0.349 | 0.000 |

[a] Grouping Variable: DW.

When compared with the average value, the variables of attractiveness, accessibility, and amenity in the tourist village group are shown to have a higher average value than

the non-tourist village group. This shows that all variables in the non-tourist village areas are lower than the tourist village areas. These findings show that tourist villages have an important role in the development of attractiveness, accessibility, and amenity.

However, even so the results of this research indicate that the establishment of a tourist village may have a positive impact on the attractiveness, accessibility, and amenity of the surrounding community. This is in accordance with some research conducted in many countries by (Agard and Roberts 2020; Chin et al. 2017; An and Alarcon 2020; Calero and Turner 2019; Ezeuduji and Rid 2011; Fun et al. 2014; Gao and Wu 2017; Tang and Jones 2012; Heijman et al. 2019; Ibanescu et al. 2018; Paresishvili et al. 2017; Pina and Delfa 2005; Ristić et al. 2019; Singgalen and Simange 2018; Teodoro et al. 2017; Trukhachev 2015; Zou et al. 2014), in which the research provided evidence strengthening of the success in increasing the rural economy through the development of rural tourism. The impact felt most from the development of a tourist village is the aspect of the public, social, and economic facilities. The existence of a tourist village that can attract tourists to visit may increase the regional income from the existing fees. The main aspect that will be the reason for tourists to visit is the aspect of providing tourism supporting facilities, such as road infrastructure, markets, minimarkets, terminals, and others. To increase the existing tourist attraction, the government will improve and increase the availability of public facilities so that the impact is that public facilities are becoming better and more complete. The better the existing facilities are, the more dense the movement of the community will be, which will provide opportunities for the opening of large markets that will open up business opportunities for the surrounding community. The opening of these business opportunities will certainly be able to increase the availability of jobs in order to reduce unemployment. The more tourists visit, the management of tourism will require additional workers.

People who have no opportunity to have business can be absorbed into the labor force at the managed tourism objects. The opening of business opportunities and the opening of jobs will be able to increase the income of the surrounding community, as explained by a research (Kurniawan 2015) that the existence of a tourist village will be able to provide a multiplier effect for the economy such as increasing the regional income, opening business opportunities, and increasing the income of the surrounding community. A study conducted by (Hamzah et al. 2018) also provided the same explanation that the impact of the existing tourist objects in an area will be able to provide greater employment opportunities for the surrounding community. Another study (Ningsih and Suryasih 2018) also explained the same thing, namely that the existence of tourist objects will be able to open business opportunities and work opportunities for the community so as to increase the income of the local community.

When viewed from the social aspect, the crowd of tourists visiting a tourist village will bring their own culture and characteristics. These culture and characteristics can be reflected in the religion adhered to so that the arrival of tourists who have different religious backgrounds requires the availability of places to worship such as mosques, temples, churches, and others. Thus, the existence of a tourist village will be able to have a positive impact on increasing inter-religious tolerance due to the availability of complete and different places to worship. In addition, the arrival of tourists who have different cultures and characteristics will also have an impact on the acculturation of culture and language so that it will enrich the local culture. A study conducted by Djabbar et al. (2021) explains that the social impact of the existing tourist objects is that local culture and customs are increasingly maintained due to increasing public awareness to maintain this culture so that it can be used as a cultural value added to tourism activities.

Regarding the achievement of Village SDGs, the development of a tourism village has proven to be able to encourage the achievement of Village SDGs. To find out whether there is a significant average difference between the variables of the tourism village group and the non-tourism village group, Table 3 presents the necessary data.

**Table 3.** Mann–Whitney U Test Result of Tourist Village and Non-Tourist Village on Output Variables.

| | Test Statistics [a] | |
|---|---|---|
| | **DVI** | **SDGs** |
| Mann–Whitney U | 1,520,712.500 | 1,419,059.500 |
| Wilcoxon W | 1,594,632.500 | 34,846,635.500 |
| Z | −1.045 | −3.187 |
| Asymp. Sig. (2-tailed) | 0.296 | 0.001 |

[a] Grouping Variable: DW.

Based on Table 3, it can be seen that of the two variables analyzed, the SDG variable has a significant average difference between the tourist village group and the non-tourist village group because the two-way (t-tailed) significance value is <0.05. By contrast, the DVI variable has no significant difference because the significance value is 0.296 > 0.05. The results of research are in line with similar research conducted in Romania by Ibanescu et al. (2018), where there are significant differences between rural areas that develop the tourism sector and those that do not develop the tourism sector. By using the same analysis, the Mann–Whitney U test, this research also demonstrates that there is a significant positive influence between villages with tourist visits and those without tourist visits, which is indicated by a higher value of all the indices analyzed.

Furthermore, in analyzing the effect of the input factors on the output factors with the status of a tourist village as a moderating variable, Partial Least Square (PLS) analysis was carried out, and the model of analysis can be seen in Figure 2. The first data validity test was carried out through a convergent validity approach, where the indicators were assessed based on the correlation between the item score/component score. Validity testing for reflective indicators can be conducted by using the correlation between the indicator score and the construct score. The measurements show that there is a change in an indicator in a construct when other indicators in the same construct change. Convergent validity can be accepted or the data declared valid if it has a loading factor value ≥ 0.5. The convergent validity test results using Smart PLS 3.3 software can be seen in Table 4 as follows:

**Table 4.** Output Result for Outer Loading.

| Variables | Indicators | Outer Loading | Description |
|---|---|---|---|
| Attractiveness | DT | 1.000 | Valid |
| Accessibility | AK1 | 0.839 | Valid |
| | AK2 | 0.922 | Valid |
| | AK3 | 0.940 | Valid |
| | AK4 | 0.908 | Valid |
| Amenity | AM1 | 0.639 | Valid |
| | AM2 | 0.566 | Valid |
| | AM3 | 0.660 | Valid |
| | AM4 | 0.845 | Valid |
| | AM5 | 0.811 | Valid |
| | AM6 | 0.801 | Valid |
| | AM7 | 0.830 | Valid |
| | AM8 | 0.797 | Valid |
| | AM9 | 0.836 | Valid |
| | AM10 | 0.879 | Valid |

**Table 4.** *Cont.*

| Variables | Indicators | Outer Loading | Description |
|---|---|---|---|
| SDGs | Y11 | 0.826 | Valid |
| | Y12 | 0.917 | Valid |
| | Y13 | 0.837 | Valid |
| | Y14 | 0.845 | Valid |
| | Y15 | 0.910 | Valid |
| | Y16 | 0.892 | Valid |
| | Y17 | 0.743 | Valid |
| | Y18 | 0.721 | Valid |
| DVI | Y21 | 0.843 | Valid |
| | Y22 | 0.887 | Valid |
| | Y23 | 0.983 | Valid |
| Tourist Village | Z | 1.000 | Valid |

Based on the table above, it can be explained that all indicators for each variable have a loading factor value above 0.5. Hence, it can be concluded that all indicators meet the requirements of convergent validity. The output loading factor in this research model is presented in Figure 4 as follows:

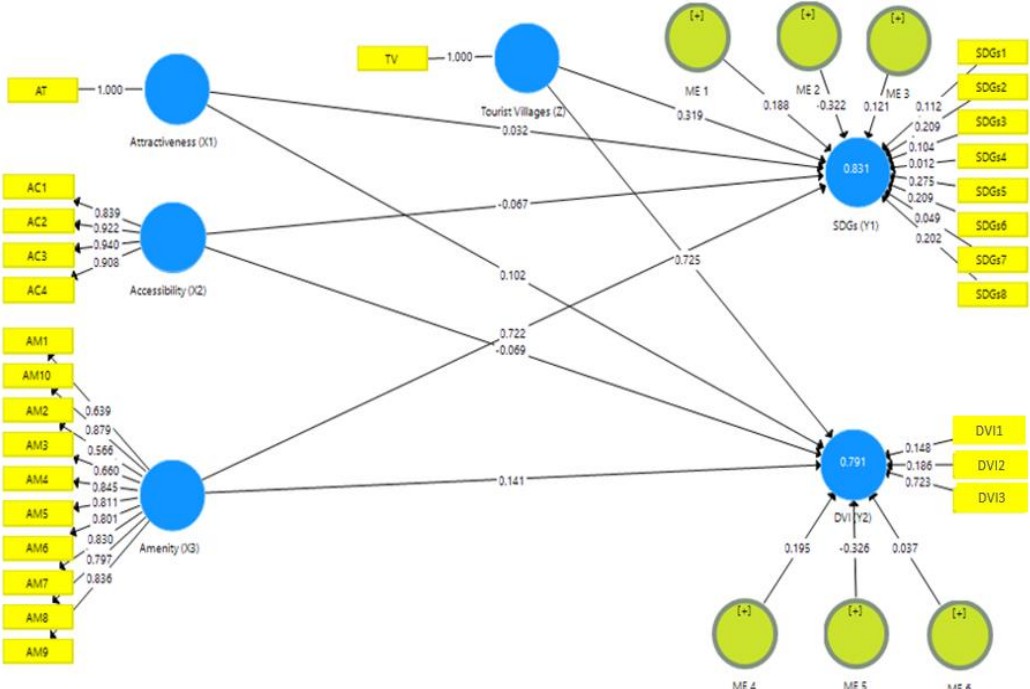

**Figure 4.** Output Loading Factor.

The next analysis is the bootstrapping test to see the significance of each relationship between the variables. The results of this analysis can be seen in Figure 5 and Table 5 as follows:

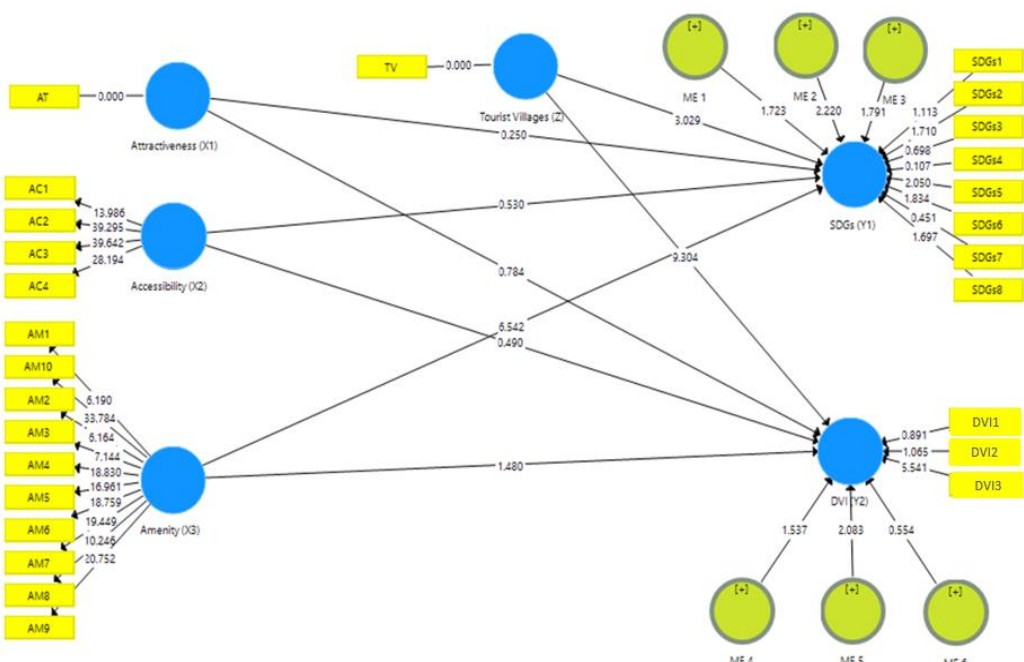

**Figure 5.** Bootstrapping Test Result.

**Table 5.** Bootstrapping Test Result.

| | Original Sample (O) | Sample Mean(M) | Standard Deviation (STDEV) | T Statistics (IO/STDEV) | *p* Values |
|---|---|---|---|---|---|
| Accessibility (X2) -> DVI (Y2) | −0.069 | −0.065 | 0.140 | 0.494 | 0.621 |
| Accessibility (X2) -> SDGs (Y1) | −0.067 | −0.083 | 0.121 | 0.556 | 0.579 |
| Amenity (X3) -> DVI (Y2) | 0.141 | 0.139 | 0.099 | 1.421 | 0.156 |
| Amenity (X3) -> SDGs (Y1) | 0.722 | 0.748 | 0.102 | 7.098 | 0.000 |
| Attractiveness (X1) -> DVI (Y2) | 0.102 | 0.092 | 0.128 | 0.796 | 0.427 |
| Attractiveness (X1) -> SDGs (Y1) | 0.032 | 0.043 | 0.123 | 0.258 | 0.796 |
| Tourist Village (Z) - > DVI (Y2) | 0.725 | 0.741 | 0.086 | 8.466 | 0.000 |
| Tourist Village (Z) -> SDGs (Y1) | 0.319 | 0.295 | 0.095 | 3.345 | 0.001 |
| Moderating Effect 1 -> SDGs (Y1) | 0.188 | 0.173 | 0.112 | 1.683 | 0.093 |
| Moderating Effect 2 -> SDGs (Y1) | −0.322 | −0.286 | 0.145 | 2.219 | 0.027 |
| Moderating Effect 3 -> SDGs (Y1) | 0.121 | 0.106 | 0.065 | 1.865 | 0.063 |
| Moderating Effect 4 -> DVI (Y2) | 0.195 | 0.191 | 0.134 | 1.453 | 0.147 |
| Moderating Effect 5 -> DVI (Y2) | −0.326 | −0.313 | 0.161 | 2.022 | 0.044 |
| Moderating Effect 6 -> DVI (Y2) | 0.037 | 0.043 | 0.063 | 0.581 | 0.561 |

From the botstraping test results presented in Figure 5 and Table 5 it can be seen:

**Hypothesis 1.** *Attractiveness has a significant effect on the SDGs.*

**Hypothesis 2.** *Attractiveness has a significant effect on DVI.*

**Hypothesis 3.** *Accessibility has a significant effect on the SDGs.*

**Hypothesis 4.** *Accessibility has a significant effect on DVI.*

**Hypothesis 5.** *Amenity has a significant effect on the SDGs.*

**Hypothesis 6.** *Amenity has a significant effect on DVI.*

**Hypothesis 7.** *Tourist Village has a significant effect on the SDGs.*

**Hypothesis 8.** *Tourist Village has a significant effect on DVI.*

**Hypothesis 9.** *Tourist Villages are significantly able to moderate the relationship between attractiveness and SDGs.*

**Hypothesis 10.** *Tourist Villages are significantly able to moderate the relationship between accessibility and SDGs.*

**Hypothesis 11.** *Tourist Villages are significantly able to moderate the relationship between amenity and SDGs.*

**Hypothesis 12.** *Tourist villages are significantly able to moderate the relationship between attractiveness and DVI.*

**Hypothesis 13.** *Tourist villages are significantly able to moderate the relationship between accessibility and DVI.*

**Hypothesis 14.** *Tourist Villages are significantly able to moderate the relationship between amenity and DVI.*

Based on the table above, it can be explained that there are five hypotheses that are accepted because they have a significant value (*p*-value <0.05). While the other hypotheses are rejected because they have a significance value (*p*-value > 0.05). Hypothesis 10 shows that the tourism village variable is able to significantly moderate the relationship between accessibility and the SDGs. These results indicate that the status of a tourist village can increase the role of accessibility in achieving the SDGs. Hypothesis 12 shows that the tourism village variable is able to significantly moderate the relationship between accessibility and DVI. These results indicate that the status of a tourist village can increase the role of accessibility in achieving DVI.

## 4. Conclusions

From the results of mapping the distribution of tourist village locations and tourist attractions in Central Java, as presented in Figure 3, it can be seen that spatially there are groupings of tourist village location points in some regions. This grouping is found in some areas in the Semarang Regency, Kudus Regency, Magelang Regency, Purworejo Regency, and Pemalang Regency. This grouping pattern shows that tourist villages are developing in regency areas that have a high enough number of villages, have rural tourist attractions, and have good infrastructure.

From the input variables (attractiveness, SDGs and DVI) analyzed, almost all variables have a significant average difference between the tourist village group and the non-tourist village group because the two-way (t-tailed) significance value is <0.05; only the accessibility variable has no significant difference because the significance value is 0.439 > 0.05. Meanwhile, the output variable, the SDGs variable, has a significant average difference between the tourist village group and the non-tourist village group because the two-way (t-tailed) significance value is <0.05, while the DVI variable has no significant difference because the significance value is 0.296 > 0.05.

The existence of tourist villages in general can have a positive impact in increasing attractiveness, amenities, and SDGs. These findings prove that the existence of a tourist village is able to support the achievement of the Village SDGs. This is in accordance with previous studies, which found that tourism development can have a positive effect on village economic conditions. In terms of accessibility, there is no significant difference between tourist villages and non-tourism villages because currently the development of accessibility infrastructure in rural areas is evenly distributed, not only in tourist villages but in all villages. As for DVI, the existence of tourist villages has not been able to encourage the achievement of DVI because there are many elements that make up DVI, namely

elements of social, economic and environmental resilience, where these elements are not only influenced by the aspects of attractiveness, accessibility and amenities.

**Author Contributions:** Conceptualization, F.F., Y.P., M.M. and S.S.; methodology, F.F. and M.M.; software, F.F.; validation, Y.P., M.M. and S.S.; formal analysis, F.F. and M.M.; investigation, F.F.; resources, F.F. and S.S.; data curation, Y.P. and M.M.; writing—original draft preparation, F.F.; writing—review and editing, F.F., Y.P., M.M. and S.S.; visualization, F.F.; supervision, Y.P., M.M. and S.S.; project administration, F.F. All authors have read and agreed to the published version of the manuscript.

**Funding:** This research received no external funding.

**Institutional Review Board Statement:** Ethical review and approval were waived for this study due to meet the ethical requirements of international research publication.

**Informed Consent Statement:** Not applicable.

**Data Availability Statement:** The corresponding author [F.F.] of the present work is available for any information about data availability.

**Conflicts of Interest:** The authors declare no conflict of interest.

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
