# Peer review of "Tourism Village Development: Measuring the Effectiveness of the Success of Village Development"

_economies, doi:10.3390/economies11050133_

Round 1
Reviewer 1 Report
It's great to have this opportunity to review this research work. The relationship between tourism and village development is always a great topic to pay attention to , because vilages have unique characteristics and different background. This work offers an interesting angle to approach to the research target. There are several suggestion to the work:
1. introduction. it's a little bit long and complex but rooms for improvement in this part. I mean, an introduction in a paper should explain background of the research, gap in the current literatures, plans of this research, and expected contribution. I suggest authors to pay more attention to polish the introduction.
2. all abbreviations when shown the first time should be explained thoroughly. SDG and IDM are used in the first part of the paper without necessary explaination, which can conduct confusion to the readers. On the contrary, some terms, like GDP, which is well-known, do not need more explanation.
3.Conclusion and discussion is the most important part in a paper, while this piece of work has simple conclusion, which should be strongly improved.
Author Response
Dear Economies Journal Editor and Reviewers
First of all, we would like to thank the editor and reviewers for their great constructive comments and helpful suggestions.
Here, we revised my draft by accommodating all of the reviewer's comments. The revision is printed in three colour (yellow for reviewer 1, green for reviewer 2, and blue for reviewer 3) in the full text.
Hopefully, this version in in line with the reviewer's suggestions.
Sincerely yours.

Reviewer 2 Report
1. What is the main question addressed by the research?
The aim of the reviewed article was to identify the distribution of tourist villages in the Central Java province using a spatial approach and to analyze the current conditions in terms of: a) comparison of conditions in tourist and non-tourist villages from input aspects (attractiveness, accessibility, convenience) supporting tourism development, b) comparison of conditions in tourist and non-tourist villages tourist and non-tourist villages as seen on the basis of the development achievements of tourist villages (SDGs - explanation of abbreviations and Developing Village Index (IDM)), and c) the impact of input factors on output factors with the status of tourist destinations as a moderating variable. The goal formulated in this way, although correct, seems to be too long, which is a certain difficulty in reading and analyzing the material.
2. Do you consider the topic original or relevant in the field? Does it
address a specific gap in the field?
The material is quite interesting, although the author did not avoid ill-considered wording, for example the author claims: "Tourism wealth will not decrease along with more foreign tourists visiting and enjoying their holidays in a country". It is difficult to fully agree with this statement, because it is well known that excessive tourist traffic has a negative impact, especially on the environment. Therefore, in selected places in the world, access to some tourist attractions is limited.
3. What does it add to the subject area compared with other published
material?
The authors try to show methods that can be used to assess the tourist attractiveness of regions (in the analyzed article of tourist villages).
4. What specific improvements should the authors consider regarding the
methodology? What further controls should be considered?
The methodological part definitely lacks a full discussion of the indicators and abbreviations used. It is not acceptable to assume that the reader must know every method, and moreover, that the reader should guess what the abbreviations used by the author mean. The lack of a proper explanation of the abbreviations makes it difficult to read the article from the very beginning. Abbreviations should be consistently explained at their first occurrence, and formulas, indicators ... in the methodological part.
5. Are the conclusions consistent with the evidence and arguments
presented and do they address the main question posed?
Conclusions are properly presented, consistent with the adopted hypotheses and properly justified.
6. Are the references appropriate?
The number of the cited bibliographic publications is appropriate and consistent with the current research issues.
7. Please include any assitional comments on the tables and figures.
I have no objections to the figures and tables.
Author Response

(The authors gave the same response as above.)

Reviewer 3 Report
Dear author,
I first of all would like to appreciate this manuscript to submit this journal. However, I found some errors and lapse in your manuscript.
1. You started the introduction suddenly from the fact of Indonesia. I suggest that you can start your introduction from the World tourism situation.
2. You can improve the manuscript by using a enough previous literature related to this title.
3. In your manuscript, I am unable to find a valid citation to your argument.
4. In my viewpoint, the methodology should be drawn in past tense. But you draw this in the present test. It should be changed.
5. In your manuscript, you did not clearly define the research model, meaning that empirical model. So, you have to please define this.
6. If you adopt the above, your manuscript will be modified in a good and acceptable form.
Therefore, I recommend this to major revision.
Author Response

(The authors gave the same response as above.)

Round 2
Reviewer 3 Report
Thank you for your correction. I recommend this manuscript with the demand of minor correction. You should consider the spellings.